# Retinal Alterations as Potential Biomarkers of Structural Brain Changes in Alzheimer’s Disease Spectrum Patients

**DOI:** 10.3390/brainsci13030460

**Published:** 2023-03-08

**Authors:** Zheqi Hu, Lianlian Wang, Dandan Zhu, Ruomeng Qin, Xiaoning Sheng, Zhihong Ke, Pengfei Shao, Hui Zhao, Yun Xu, Feng Bai

**Affiliations:** 1Department of Neurology, Nanjing Drum Tower Hospital, The Affiliated Hospital of Nanjing University Medical School, and The State Key Laboratory of Pharmaceutical Biotechnology, Institute of Brain Science, Nanjing University, Nanjing 210008, China; 2Department of Neurology, Nanjing Drum Tower Hospital Clinical College of Jiangsu University, Nanjing 210008, China; 3Department of Ophthalmology, Nanjing Drum Tower Hospital, The Affiliated Hospital of Nanjing University Medical School, Nanjing University, Nanjing 210008, China; 4Geriatric Medicine Center, Affiliated Taikang Xianlin Drum Tower Hospital, Medical School of Nanjing University, Nanjing 210008, China

**Keywords:** Alzheimer’s disease, biomarker, cognitive impairment, optical coherence tomography angiography, vessel density, retinal nerve fiber layer, hippocampus subfields, diffusion tensor imaging, white matter integrity

## Abstract

Retinal imaging being a potential biomarker for Alzheimer’s disease is gradually attracting the attention of researchers. However, the association between retinal parameters and AD neuroimaging biomarkers, particularly structural changes, is still unclear. In this cross-sectional study, we recruited 25 cognitively impaired (CI) and 21 cognitively normal (CN) individuals. All subjects underwent retinal layer thickness and microvascular measurements with optical coherence tomography angiography (OCTA). Gray matter and white matter (WM) data such as T1-weighted magnetic resonance imaging and diffusion tensor imaging, respectively, were also collected. In addition, hippocampal subfield volumes and WM tract microstructural alterations were investigated as classical AD neuroimaging biomarkers. The microvascular and retinal features and their correlation with brain structural imaging markers were further analyzed. We observed a reduction in vessel density (VD) at the inferior outer (IO) sector (*p* = 0.049), atrophy in hippocampal subfield volumes, such as the subiculum (*p* = 0.012), presubiculum (*p* = 0.015), molecular_layer_HP (*p* = 0.033), GC-ML-DG (*p* = 0.043) and whole hippocampus (*p* = 0.033) in CI patients. Altered microstructural integrity of WM tracts in CI patients was also discovered in the cingulum hippocampal part (CgH). Importantly, we detected significant associations between retinal VD and gray matter volumes of the hippocampal subfield in CI patients. These findings suggested that the retinal microvascular measures acquired by OCTA may be markers for the early prediction of AD-related structural brain changes.

## 1. Introduction

Alzheimer’s disease (AD) is the most prevalent form of dementia among elderly individuals worldwide, affecting over 50 million people [1]. The pathophysiological alterations in AD involve the loss of neurons, brain atrophy, the deposition of extracellular β-amyloid (Aβ) plaques and the accumulation of intracellular neurofibrillary tangles. Early diagnosis of AD is critical for effective prevention and management. Mild cognitive impairment (MCI) is defined as a transitional phase between normal aging and AD, and patients with MCI have a 10–15% higher risk of developing AD per year [2]. Established markers for the diagnosis of AD include cerebrospinal fluid markers as well as imaging biomarkers such as magnetic resonance imaging (MRI) and positron emission tomography (PET). Brain structural decline offers a versatile means of understanding and targeting pathophysiological mechanisms [3].

The retina is considered an extension of the central nervous system and neurodegenerative changes resulting from diseases are mirrored in the eye [4]. At the molecular level, several studies have demonstrated the presence of classical AD biomarkers, such as amyloid β and phosphorylated tau, within the retinal structure and vasculature. These deposits cause retinal and capillary degeneration, such as apoptosis of retinal ganglion cells (RGCs), thinning of the retinal nerve fiber layer (RNFL) and other structural and functional impairments in AD patients [5,6,7,8,9,10,11,12,13,14,15,16,17,18]. Optical coherence tomography (OCT) is an optical imaging technique that enables both structural and functional imaging of the retina in humans, as well as a wide range of veterinary animals [19,20,21]. OCT angiography (OCTA) allows for the extraction of blood vessels in the retina from OCT data. Various reviews have highlighted characteristic pathological retinal changes in AD patients, including a significant reduction in RNFL thickness and macular and choroidal thickness, a decline in vessel density and an increase in the foveal avascular zone (FAZ) in patients with AD, even in MCI subjects, as measured by OCTA [22,23,24,25,26]. Furthermore, some studies have reported an association between a thinner retinal structure and worse cognitive function, as well as a greater likelihood of future cognitive decline [27,28,29].

The relationship between changes in retinal structure or microvasculature and the neuroimaging of Alzheimer’s disease (AD) has received increasing attention in recent studies. Previous research has shown that a thinner macular thickness is negatively correlated with cerebral cortical atrophy, particularly parietal cortical atrophy in AD patients [30,31]. Moreover, a thicker retinal nerve fiber layer (RNFL) has been linked to better MRI variables, such as greater hippocampal volumes and improved diffusion tensor imaging variables in elderly individuals without dementia [32,33,34]. In an OCTA study, the authors found that reductions in retinal vessel density (VD) were significantly associated with inferolateral ventricle (ILV) enlargement in MCI and AD. However, most of these studies focused on the hippocampus in large scale measures despite its heterogeneous composition consisting of distinct subfields with varying anatomical, functional and electrophysiological properties, including the dentate gyrus, subiculum, parasubiculum and presubiculum [32,35,36,37,38]. Multiple studies had reported that atrophy in the CA1, subiculum and presubiculum represent the earliest sites for AD pathologic changes (i.e., amyloid deposits, tau aggregation) [39,40,41]. Additionally, diffusion tensor imaging (DTI) is the most common technique to assess pathophysiology in neurodegenerative diseases, especially the detection of microstructural changes in the cerebral white matter (WM) in AD. A multicenter study with a large cohort found a significantly decreased fractional anisotropy (FA) and an increased mean diffusivity (MD) in areas with AD pathology, including the corpus callosum, cingulate gyrus, fornix, precuneus, medial and lateral temporal lobes and prefrontal lobe WM [42]. Previous studies also showed the potential correlation between retinal layer thickness changes and WM variables [32,43,44].

Here, we aimed to investigate the changes in retinal measures between groups and determine the correlation between retinal layer thickness/microvascular measures and classical AD structural biomarkers in cognitively normal (CN) subjects and cognitively impaired (CI) patients. Our hypothesis was that retinal characteristics can reflect altered AD-related structural brain changes, providing a simple and non-invasive biomarker with potential clinical applications.

## 2. Materials and Methods

### 2.1. Participants

The present study recruited 52 right-handed Chinese Han elderly individuals at the Department of Neurology of the Nanjing Drum Tower Hospital of Nanjing University Medical School from September 2020 to January 2021, including 10 AD subjects (4 males and 6 females), 21 MCI subjects (6 males and 15 females) and 21 CN individuals (7 males and 14 females). This study was approved by the ethics committees of the Nanjing Drum Tower Hospital and Nanjing University Medical School, and written informed consent was obtained from each subject prior to participation. All subjects underwent a comprehensive neuropsychological test, 3.0 T whole brain MRI scanning and a general medical examination by an experienced neurologist. The cognitive functions of all the subjects were evaluated by an experienced neuropsychologist using the Chinese version of the Mini-Mental State Examination (MMSE) and the Beijing version of the Montreal Cognitive Assessment (MoCA) as general cognitive function screening. The Clinical Dementia Rating (CDR) scale, activities of daily living (ADL) assessment, Hamilton Depression Rating Scale (HAMD) and Hamilton Anxiety Rating Scale (HAMA) were also tested.

The inclusion criteria for CN subjects were as follows: (1) the absence of reported cognitive complaints; (2) normal MMSE and MoCA scores that were adjusted for age and education. The MoCA was used to identify CN and CI. Optimal cutoff points were determined based on education level. For individuals with 7 or more years of education, the MoCA cutoff was 24/25. For those with 1–6 years of education, the MoCA cutoff was 19/20. Finally, for those with no formal education, the cutoff was 13/14; and (3) CDR = 0 [45,46]. These CI patients included MCI and AD patients. In contrast, the inclusion criteria for CI patients were determined using the National Institute of Neurological and Communicative Disorders and Stroke and the AD and Related Disorders Association (NINCDS-ADRDA) criteria for AD patients and the Petersen criteria for MCI patients [47,48]: (1) subjective memory complaint confirmed by an informant; (2) objective memory impairment as detected by the MoCA or Auditory Verbal Learning Test (AVLT) scores that were at least 1.5 standard deviations below normative values for age and/or education; (3) preserved general cognitive function (MMSE ≥ 24); (4) CDR score = 0.5; (5) no or minimal impairment in activities of daily living (ADL); and (6) not sufficient to diagnose dementia. Exclusion criteria for all the subjects were as follows: (1) age less than 50 years old; (2) a history of stroke, diabetes or hypertension; (3) central nervous system diseases that could potentially result in cognitive decline; (4) severe mental health conditions such as schizophrenia, anxiety (Hamilton Rating Scale for Anxiety (HAMA) score ≥ 21) or depression (Hamilton Rating Scale for Depression (HAMD) score ≥ 17); (5) severe systemic diseases such as heart failure and kidney dysfunction; (6) a history of drug or alcohol abuse; (7) intolerance of MRI examination or the inability to complete neuropsychological testing; (8) other medical conditions that may impact cognition; and (9) the presence of concurrent retinal diseases such as diabetic neuropathy, epiretinal membrane or macular degeneration, as well as any history of glaucoma, optic neuropathies or ocular surgery, except for cataract surgery. Additionally, all participants underwent thorough ophthalmic evaluations, with eyes having a history of retinal surgery, evidence of epiretinal membrane, glaucoma or branch retinal vein occlusion being excluded. A flowchart (Figure 1) visually represented the three main processing sectors of the study, which will be discussed in further detail. Finally, a total of 46 subjects (90 eyes) were enrolled in this study, including 25 CI patients (17 MCI patients, 8 AD patients) and 21 CN subjects (Appendix A). 

### 2.2. MRI Data Acquisition

In this study, imaging data were acquired using a 3.0T scanner, specifically the 3.0T Ingenia (32-channel head coil), manufactured by Philips in Eindhoven, Netherlands, which was located at Nanjing Drum Tower Hospital. To obtain sagittal T1-weighted MR images covering the entire brain, participants were positioned in the supine position and underwent a three-dimensional turbo fast echo acquisition. The T1-weighted imaging acquisition parameters were: a repetition time (TR) of 8.2 ms, an echo time (TE) of 3.7 ms, a field of view (FOV) of 256 × 256 mm, an acquisition matrix of 256 × 256, a flip angle (FA) of 8°, a slice thickness of 1.0 mm, no gap, voxel resolution  =  1 × 1 × 1 mm^3^ and a number of slices of 192. Diffusion-weighted imaging was performed using an echo planar imaging (EPI) sequence, with diffusion-sensitizing gradients applied along 32 non-collinear directions (b = 1000 s/mm^2^), and an acquisition without diffusion weighting (b = 0). The acquisition parameters for diffusion-weighted imaging were: a TR of 8500 ms, a TE of 71 ms, a flip angle of 90°, a matrix size of 112 × 112 and an FOV of 224 × 224 mm. In addition, an axial T2-weighted, diffusion-weighted imaging (DWI) sequence and fluid-attenuated inversion recovery (FLAIR) sequence were acquired to detect acute or subacute infarctions and visible white matter damage.

### 2.3. MRI Analysis

#### 2.3.1. Hippocampal Subfield Acquisition

The present study employed the FreeSurfer software (version 6.0) and the integrated hippocampal subfield segmentation software package to perform hippocampal subfield segmentation and grey/white matter volumetric segmentation [49]. The standard FreeSurfer processing pipeline using the “recon-all” script was utilized to preprocess all T1-weighted images [50]. The volumetric segmentation process for hippocampal subfields, which has been described previously, involved dividing a total of 12 subfields for each side of the hippocampus: the parasubiculum, presubiculum, subiculum, CA1, CA2-CA3, CA4, granule cell layer of the dentate gyrus, hippocampus–amygdala transition area, fimbria, molecular layer, hippocampal fissure and hippocampal tail (Figure 2). However, because the fimbria (white matter) and hippocampal fissure (cerebrospinal fluid) are not part of the grey matter and have shown relatively lower segmentation accuracies than other subfields in previous research, they were excluded from the subsequent analysis [51,52]. To ensure accuracy, a quality control process and manual editing were implemented and poor quality of segmentation were excluded [53].

#### 2.3.2. DTI Processing

In this study, the Atlas-based segmentation approach was utilized to investigate diffusion abnormalities. DTI data processing was conducted using PANDA software (http://www.nitrc.org/projects/panda/, accessed on 12 February 2021), following the default pipeline setting [54]. PANDA is a pipeline tool that integrates FSL, the Diffusion Toolkit and MRIcron for diffusion MRI analysis. The data processing steps included converting the original DICOM data to a NIFIT format, removing non-brain tissue, correcting for eddy current and head motion, adjusting the diffusion gradient direction and calculating the diffusion tensor metrics, such as fractional anisotropy (FA), mean diffusivity (MD), radial diffusivity (RD) and axial diffusivity (AxD). The FA images in native space were registered to the FA standard template in Montreal Neurological Institute (MNI) space using FSL’s FNIRT command. To evaluate changes in the major white matter tracts, all DTI metrics were registered to the JHU White Matter Tractography Atlas [55]. In the present research, we obtained FA/MD/AxD/RD diffusion tensor metrics.

Specific calculation indexes are as follows: (1)$$\mathrm{FA}=\sqrt{3{\left(\lambda_{1}-\bar{\lambda }\right)}^{2}+{\left(\lambda_{2}-\bar{\lambda }\right)}^{2}+{\left(\lambda_{3}-\bar{\lambda }\right)}^{2}}/\sqrt{2(\lambda_{1}^{2}+\lambda_{2}^{2}+\lambda_{3}^{2})}$$

MD = (λ_1_ +λ_2_ + λ_3_)/3; AxD = λ_1_; RD = (λ_1_ + λ_3_)/2. In this study, the diffusion of water molecules was characterized by three dispersion directions, namely λ_1_, λ_2_ and λ_3_ [56]. Specifically, λ_1_ indicates the axial direction of the fiber bundle within the voxel, while λ_2_ and λ_3_ represent the radial directions perpendicular to the axis. The WM fiber pathways of interest included anterior thalamic radiation (ATR); cingulum in the cingulated cortex area (CgC); cingulum in the hippocampal area (CgH); corticospinal tract (CST); forceps major (FMa); inferior fronto-occipital fasciculus (IFO); inferior longitudinal fasciculus (ILF); superior longitudinal fasciculus (SLF); temporal projection of the SLF (tSLF); and uncinate fasciculus (UF). Bilateral evaluations were conducted for all tracts, with the exception of the corpus callosum (forceps major) and corpus callosum (forceps minor).

### 2.4. Retinal Image Acquisition

Retinal imaging was carried out with a spectral-domain OCTA machine (Cirrus HD-5000 AngioPlex; Carl Zeiss Meditec, Dublin, CA, USA) capable of scanning at 68,000 A-scans/s. Images of 6 × 6 mm centered on the fovea, 512 × 128 for the macular cube and 200 × 200 for the optic disc cube scans were captured. Images that were poor quality (less than 7/10 signal strength) or had low resolution, uncorrectable segmentation errors, projection artifacts or motion artifacts were excluded. Segmentation of full-thickness retinal scans in the superficial capillary plexus (SCP) was automated using OCTA software (Version 10.0; Carl Zeiss Meditec, Dublin, CA, USA). The software automatically quantified the average VD for the 6 × 6 mm SCP images over the central macula using an Early Treatment Diabetic Retinopathy Study (ETDRS) grid overlay, and it was automatically calculated for the 6 mm circle, 6 mm ring and 3 mm ring regions of the 6 × 6 mm OCTA images (Appendix A). The software also automatically segmented and quantified the FAZ area. RNFL thickness (using a 3.46-mm diameter circle centered on the optic disc) was recorded [57,58].

### 2.5. Statistical Analysis

The measurement data were presented as mean ± standard deviation (SD). Independent sample t-test and Mann–Whitney U test were used to compare normally and non-normally distributed measurement data between groups, respectively. The χ^2^ test was employed to examine between-group sex differences, which is categorical data. All *p* values were two-sided, and no correction was made for multiple analyses. To minimize Type 1 errors caused by conducting multiple tests, we averaged all retinal measures, DTI metrics and hippocampal subfield volumes for both hemispheres. We performed an analysis of covariance (ANCOVA) to investigate whether retinal measures, hippocampal subfield volumes or alterations in WM integrity were changed in CI participants, while adjusting for sex, age, years of education and estimated Total Intracranial Volume (eTIV) [59]. To evaluate the linear correlation between the averaged retinal parameters in the optic disc and macula with AD neuroimaging biomarkers, partial correlation analysis was used, with age, gender, education and eTIV acting as calibration control variables. All statistical analyses were performed using SPSS for Windows (version 26.0, IBM, Chicago, IL, USA). Statistical significance was considered at *p* < 0.05 (two-tailed).

## 3. Results

### 3.1. Demographic, Neuropsychological Characteristics and Retinal Measures

The demographic and clinical data of the CN and CI groups are presented in Table 1. No significant differences were observed between the groups with regard to gender (*p* = 0.924), eTIV (*p* = 0.526), HAMD (*p* = 0.58) or HAMA scores (*p* = 0.74). However, the individuals in the CI group were found to be significantly older than those in the CN group (*p* = 0.016) and had a lower level of education (*p* < 0.001). The CI group exhibited lower scores for MMSE and MoCA-BJ assessments, in comparison to the CN group. The difference in retinal measures between the CI and CN groups is shown (Appendix A, Figure 3). The results show a decrease in vessel density (VD) at the inferior outer (IO) sector in the CI group when compared with the CN group (*p* = 0.049). RNFL and macular thickness in any sectors or rings did not discriminate CI patients from CN subjects. Trends of lower retinal VD and thinner RNFL and macular thickness in the CI group were found (Appendix A).

### 3.2. Hippocampal Volume and Its Association with Retinal Measures

In the comparison of hippocampal subfield volumes between these two groups (Table 2, Figure 4), volumes were significantly decreased in the CI group compared with the CN group in the subiculum (*p* = 0.012), presubiculum (*p* = 0.015), molecular_layer_HP (*p* = 0.033), GC-ML-DG (*p* = 0.043) and the whole hippocampus (*p* = 0.033). In addition, the CI group showed a trend of atrophy in other hippocampal subfield volumes when compared with the CN group, but the differences were not statistically significant.

To test our major hypothesis, we employed hippocampal subfields that have demonstrated significant differences previously as proxies for AD disease severity. We determined the association between retinal parameters and hippocampal subfield volumes, controlling for age, gender, education and eTIV (Appendix A). As hypothesized, we found structural and vascular retinal degeneration mirror atrophy of the hippocampal subfield volumes. As shown in Figure 5, the volumes of the subiculum were positively associated with VD_TO (r = 0.523, *p* = 0.018, Figure 5A). When comparing the presubiculum with retinal measures, we found consistent moderate to strong positive associations between the presubiculum and VD_TO (r = 0.593, *p* = 0.006, Figure 5B) and VD_IO (r = 0.559, *p* = 0.01, Figure 5F). Similarly, when investigating the association between retinal vascular changes and hippocampal subfield volumes, we found that a higher VD in the TO sector was associated with greater molecular_layer_HP (r = 0.463, *p* = 0.04) and GC-ML-DG (r = 0.452, *p* = 0.046) and whole hippocampus (r = 0.49, *p* = 0.029) volumes (Figure 5C–E). It is worth noting that there were no significant correlations between hippocampal subfield volumes and macular or RNFL OCT parameters in the CI group (all *p* > 0.05, Appendix A). 

### 3.3. WM Integrity and Its Association with Retinal Measures

We performed an analysis to estimate the WM integrity for all fiber bundles in both groups. The findings revealed a significant difference in the MD and AxD of the cingulum in the hippocampal area (CgH) (*p* = 0.046 and *p* = 0.03, respectively) between the CI and CN groups (Table 3, Figure 6, Appendix A).

In general, increased MD values and decreased FA values are indicative of compromised fiber integrity, which may result from increased diffusion in the major fiber directions and a loss of fiber myelination. Thus, we proceeded to investigate the potential association between retinal parameters and WM integrity in fibers that showed significant damage in the CI group, while controlling for age, gender and years of education. However, in the present study, we did not observe any significant correlation in the CI group (see Appendix A). Conversely, in the CN group, we found a positive correlation between the axial diffusivity (AxD) value of the cingulum in the hippocampal area (CgH) and the VD_TI retina, with a correlation coefficient of 0.495 and a *p*-value of 0.037 (Appendix A).

## 4. Discussion

Our cross-sectional study investigated the relationship between retinal layer thickness/microvascular measures and classical AD-related structural biomarkers. Results found that (i) in the CI group, a statistically significant decrease in vessel density (VD) was observed in the inferior outer (IO) sector when compared to the CN group. While the RNFL and macular thickness did not differentiate between the CI and CN groups, a decreasing trend was noted in our study. Retinal changes, as an extension of the brain, may reflect pathophysiological processes in the central nervous system, as previously demonstrated in other studies [4,22,23,24,60,61]. (ii) The present CI patients showed atrophy in hippocampal subfield volumes and an altered integrity of the microstructure of WM tracts. Importantly, we found significant associations between retinal parameters and hippocampal subfield volumes in the CI group. 

### 4.1. Volumetric Comparisons of the Hippocampal Subfield Volumes and Correlations with Retinal OCTA Parameters

In contrast to our expectations, widespread hippocampal subfield losses in CI were found, which contained subiculum, presubiculum, molecular_layer_HP and GC-ML-DG. This is consistent with a previous study that indicated CA1, subiculum, presubiculum, molecular layer and fimbria showed a trend toward significant volume reduction in the progression of AD [62]. Early research described the relation between retinal layers and MRI features in AD spectrum populations. One study found an inverse correlation between macular thickness and cerebral cortical atrophy, especially parietal cortical atrophy in AD patients [30]. Others revealed a significant positive correlation between RNFL thickness and hippocampal volume, with this association being more pronounced in the m-RNFL in people without dementia [35,36,37]. However, these studies were focused on the relationship between retinal structure and a relatively large area of the brain (i.e., temporal lobe, whole hippocampus) [35,36,37,63]. Instead, we focused on the association between retinal measures and hippocampal subfields. We found that thicker RNFL in the special sector showed significantly increased subiculum, presubiculum, GC-ML-DG and molecular_layer_HP volumes, which is consistent with previous studies [35,36,37].

Growing evidence from epidemiological, clinical and experimental studies suggests that cerebromicrovascular and microcirculatory damage related to aging play crucial roles in the development of various forms of dementia, including Alzheimer’s disease (AD), in the elderly population. The significance of microvascular contributions to AD in the elderly cannot be overstated [64]. Due to the resemblance of retinal arterioles and venules to cerebral small blood vessels, changes in retinal microvasculature may indicate similar pathological processes occurring in the cerebral microvasculature. Therefore, we also explored the association between hippocampal subfield atrophy and retinal microvasculature and discovered that the volumes of the whole hippocampus and subfields (i.e., subiculum, presubiculum, GC-ML-DG and molecular_layer_HP) were positively correlated with retinal VD. One recent study indicated a positive correlation between retinal VD and some cognitive function domains [28]. Another study found that larger presubiculum, subiculum and CA4/dentate gyrus volumes were associated with higher delayed recall scores and fewer informant reports of memory difficulties [38]. Combined with our results, special hippocampal subfields may play a mediating role in the association between retinal VD and cognitive function domains. However, no association between macular thickness or RNFL and hippocampal subfield volumes was observed. This may be because previous studies measured macular ganglion cell-inner plexiform layer (GCL-IPL) thickness; in contrast, our software automatically assessed the thickness of the ILM-RPE. From the retinal structure, we can conclude that GCL-IPL is included in the ILM-RPE and that ILM thickness is less sensitive to changes during AD progression than GCs, which may lead to such results.

In addition, we further explored whether this relationship also existed in healthy subjects and results showed a positive correlation between retinal VD and the volumes of the presubiculum (Appendix A), which is consistent with previous findings [33,63]. The findings have shown that this association exists at different stages of AD and becomes more pronounced as the disease progresses; therefore, the retina is becoming a potential marker for predicting the degree of AD progression. 

### 4.2. Microstructural Comparisons of WM Integrity and Correlations with Retinal OCTA Parameters

Diffusion tensor imaging (DTI) is a widely used technique in diffusion MRI for examining the pathophysiology of neurodegenerative disorders, including AD, Parkinson’s disease (PD) and multiple sclerosis [65]. This technique utilizes four metrics, including FA, MD, AxD and RD, to describe the brain’s structure based on water diffusion [66]. In this study, we investigated the relationship between DTI metrics and retinal parameters. Compared to normal elderly adults, patients with CI displayed elevated MD/AxD values in the CgH tracts, which are part of the limbic system and are known to be affected early in AD. Previous DTI research has shown that WM changes in AD patients primarily affect the medial temporal limbic-associated tracts, which is consistent with the disease’s pathological involvement of the temporal lobe [67,68]. Consistent with previous studies, our findings indicated decreased WM integrity in the bilateral limbic tracts (CgH) of CI patients [69,70].

Few studies have investigated the associations between WM microstructure and retinal measures. One previous study found that thicker RNFL was associated with better MRI variables in both the cingulum and posterior thalamic radiations in elderly individuals without dementia [32]. Another population-based study revealed that a thinner RNFL and ganglion cell layer (GCL) was associated with lower FA and higher MD in WM tracts, which are part of optic radiation [44]. A study exploring whether ganglion cell-inner plexiform layer (GC-IPL) thickness is associated with brain WM microstructure and how this association differs between normal and cognitively impaired subjects found that thinner GCL-IPL was linked with lower WM microstructure integrity in subjects without cognitive impairment, suggesting a possible physiological relationship between the retina and the brain in healthy aging [43]. Our study also investigated the association between retinal measures, including microvascular, macular, and RNFL and WM alterations. Interestingly, no significant correlation was found in the CI group, but a correlation existed in the CN group. We speculate that this contradictory result may be due to the limited sample size and the disruption of retina–brain associations in the presence of cognitive impairment [43].

## 5. Limitations

Our study has several limitations that should be taken into account. Firstly, the sample size was relatively small, and the data was collected cross-sectionally, preventing us from determining the temporal sequence and directionality of the observed associations. Initiating longitudinal studies with larger cohorts, including individuals at various stages of cognitive decline, and beginning retinal measurements at an early age are necessary steps to investigate the potential direct relationship between retinal and brain degeneration. Secondly, our study lacked amyloid imaging data and CSF biomarkers, which are proposed by the AT(N) biological framework [71]. Early brain Aβ pathology may be a more sensitive biomarker of AD processes, and investigating the relationship between retinal measures and Aβ levels could provide insight into the potential utility of retinal imaging as a preclinical AD biomarker [72]. Thirdly, although the retina is composed of many different structures, our measurement of macular thickness only considered the full layer thickness. In future studies, we plan to analyze the retina in greater detail, including the structure of the ONL.

## 6. Conclusions

In this study, we found a significant decrease in vessel density and associations between retinal parameters and atrophy of the hippocampal subfield volumes in CI patients. These findings suggested that retinal layer thickness and microvascular measures detected by OCTA may be markers for the early prediction of AD-related structural brain changes.

## Figures and Tables

**Figure 1 brainsci-13-00460-f001:**
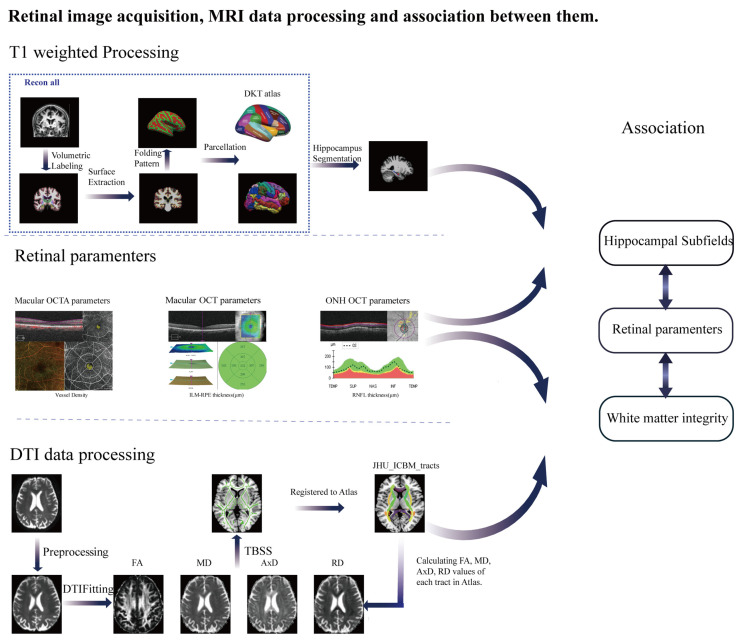
A flowchart shows the overview of the retinal imaging cohort that included retinal vessel density (VD) and thickness of the nerve fiber layer and macula measurements using OCTA (optical coherence tomography angiography) imaging and the whole pipeline of T1-weighted and DTI image processing. ILM-RPE, inner limiting membrane–retinal pigment epithelium; ONH, optic nerve head; RNFL, retinal nerve fiber layer; DTI, diffusion tensor imaging; FA, fractional anisotropy; MD, mean diffusivity; AxD, axial diffusivity; RD, radial diffusivity. TBSS, Tract-based spatial statistics.

**Figure 2 brainsci-13-00460-f002:**
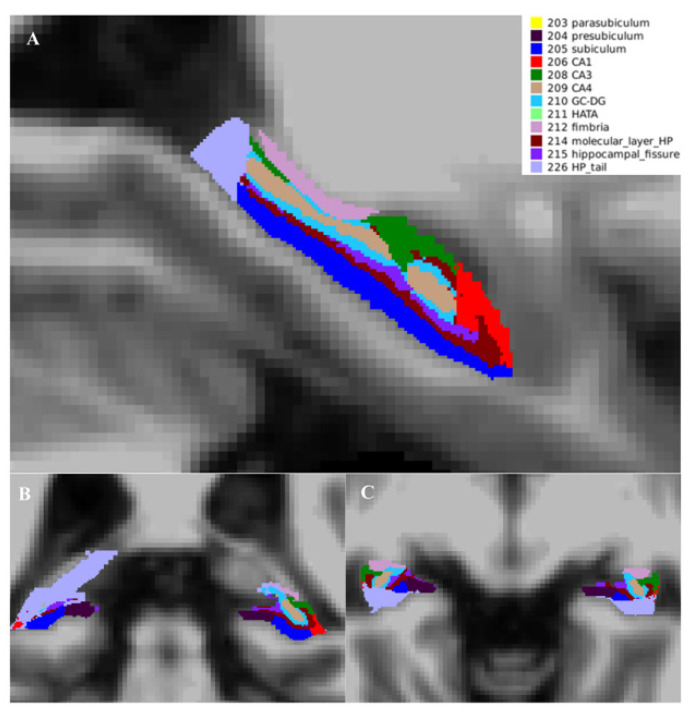
(**A**–**C**) Representative example of hippocampal subfield segmentations from a patient with cognitive impairment from sagittal, transverse and coronal views.

**Figure 3 brainsci-13-00460-f003:**
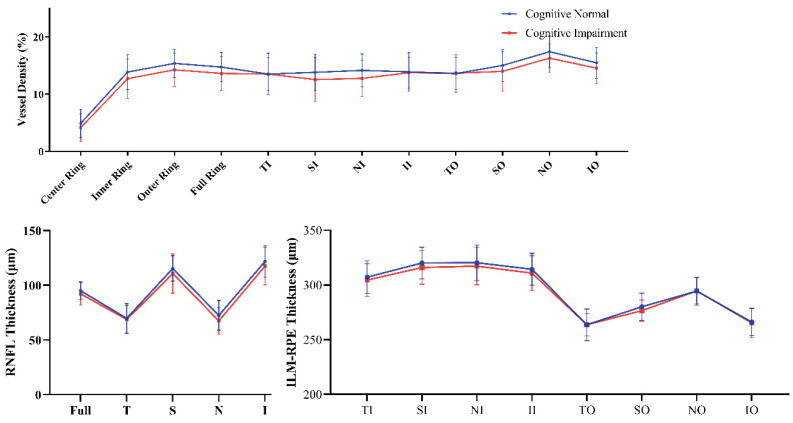
Comparison of retinal measures in subjects with cognitive impairment (CI) and cognitively normal (CN) subjects. All data were analyzed by covariance (ANCOVA), accounting for sex, age and years of education. T, temporal; S, superior; N, nasal; I, inferior; TI, temporal inner; TO, temporal outer; NI, nasal inner; NO, nasal outer; SI, superior inner; SO, superior outer, II, inferior inner, IO, inferior outer; VD, vessel density; RNFL, retinal nerve fiber layer; ILM-RPE, inner limiting membrane–retinal pigment epithelium.

**Figure 4 brainsci-13-00460-f004:**
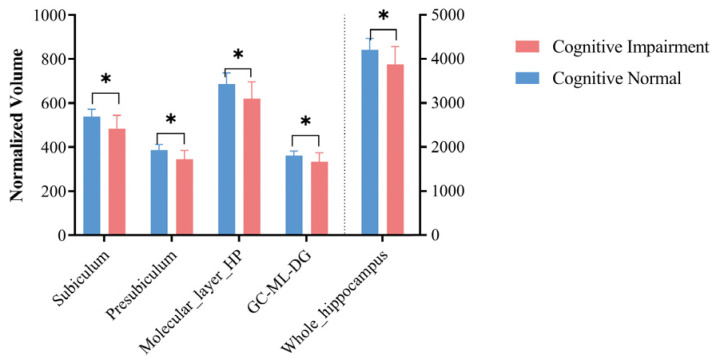
Comparison of hippocampal subfields between the cognitive impairment (CI) and cognitively normal (CN) groups. In the hippocampal subfields, compared with the CN group, the CI group showed decreased volumes in the subiculum (*p* = 0.012), presubiculum (*p* = 0.015), molecular layer HP (*p* = 0.033), GC-ML-DG (*p* = 0.043) and the whole hippocampus (*p* = 0.033). GC-ML-DG: Molecular and Granule Cell Layers of the Dentate; HATA: Hippocampal–Amygdaloid Transition Area. eTIV: estimated Total Intracranial Volume. * *p* < 0.05.

**Figure 5 brainsci-13-00460-f005:**
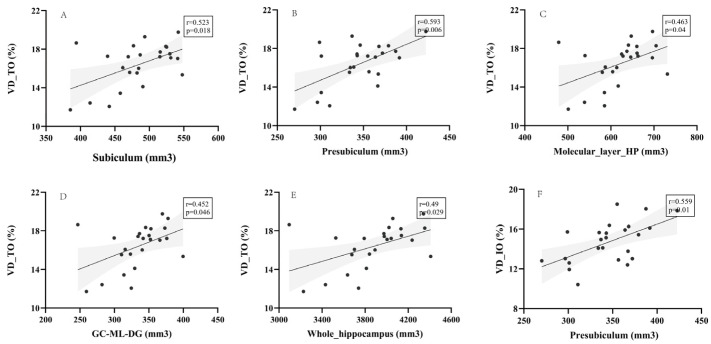
Association between retinal parameters and hippocampal subfield volumes in the cognitive impairment (CI) group. Partial correlation was conducted, controlling for age, gender, years of education and eTIV. (**A**) Subiculum volumes were positively correlated with VD_TO (r = 0.523, *p* = 0.018). (**B**,**F**) Presubiculum volumes were positively related to VD_TO (r = 0.593, *p* = 0.006) and VD_IO (r = 0.559, *p* = 0.01). (**C**–**E**) VD in the TO sector was associated with greater molecular_layer_HP (r = 0.463, *p* = 0.04) and GC-ML-DG (r = 0.452, *p* = 0.046) and whole hippocampus (r = 0.49, *p* = 0.029) volumes. VD, vessel density; GC-ML-DG: Molecular and Granule Cell Layers of the Dentate; TO, temporal outer; IO, inferior outer.

**Figure 6 brainsci-13-00460-f006:**
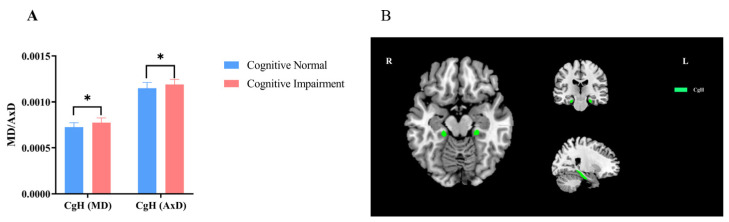
Significantly damaged fiber tracts and their locations, comparing white matter (WM) integrity between cognitive impairment (CI) and cognitively normal (CN) groups (**A**,**B**). (**A**) For WM integrity, compared with the CN group (n = 21, blue), the CI group (n = 25, red) showed increased MD and AxD values in the CgH of the CI group (uncorrected, *p* < 0.05). (**B**) Showed the location of GgH in brain regions. CgH, cingulum hippocampus; AxD, axial diffusivity; MD, mean diffusivity; R, right; L; left. * *p* < 0.05.

**Table 1 brainsci-13-00460-t001:** Demographic and neuropsychological characteristics.

Items	CN (21)	CI (24)	F/T	*p*
Age, year	61.43 ± 7.52	67 ± 7.83	−2.49	0.016 *
Education, year	13.95 ± 2.16	10.63 ± 2.5	4.92	*p* < 0.001 ***
Gender, female/male	14/7	17/8	0.009	0.924
MMSE	28.81 ± 1.17	24.5 ± 5.27	−3.83	*p* < 0.001 ***
MoCA	26.95 ± 1.5	19 ± 4.95	−5.74	*p* < 0.001 ***
eTIV	1,923,091.48 ± 145,178.78	1,987,143.36 ± 93,560.77	−0.64	0.526
HAMD	4.33 ± 4.69	5.19 ± 4.92	−0.55	0.58
HAMA	7.00 ± 6.89	7.76 ± 7.04	−0.34	0.74

The data are reported as mean ± standard deviation (SD). Age and eTIV were analyzed using independent sample t-tests, while gender was analyzed using the χ^2^ test. Education, MMSE, MoCA, HAMD and HAMA scores were analyzed using the Mann–Whitney U test. CN: cognitively normal; CI: cognitive impairment; MMSE: Mini-Mental State Examination; MoCA-BJ: Beijing version of the Montreal Cognitive Assessment; eTIV: estimated Total Intracranial Volume; HAMD: Hamilton depression scale; HAMA: Hamilton anxiety scale. * *p* < 0.05, *** *p* < 0.001.

**Table 2 brainsci-13-00460-t002:** Comparison of hippocampal subfields between the cognitively normal group and cognitive impairment group.

Items	CN (21)	CI (27)	t	*p*-Value
Hippocampal_tail	702.9 ± 61.72	663.69 ± 65.34	2.36	0.132
subiculum	539.8 ± 32.33	490.49 ± 53.87	6.99	0.012 *
CA1	744.04 ± 61.14	700 ± 62.6	3.06	0.088
hippocampal-fissure	235.78 ± 31.42	236.55 ± 35.79	0.4	0.532
presubiculum	385.86 ± 26.55	348.25 ± 37.25	6.4	0.015 *
parasubiculum	71.91 ± 8.04	75.5 ± 10.77	2.13	0.152
molecular_layer_HP	685.99 ± 50.79	626.81 ± 70.51	4.88	0.033 *
GC-ML-DG	362.08 ± 20.41	337.89 ± 37.77	4.36	0.043 *
CA3	251.61 ± 21.46	237 ± 29.88	1.98	0.167
CA4	307.47 ± 14.48	293.5 ± 31.51	3.72	0.061
fimbria	89.75 ± 10.94	79.65 ± 16.77	2.5	0.122
HATA	69.16 ± 9.77	66.39 ± 11.32	0.006	0.941
Whole_hippocampus	4210.56 ± 256.59	3919.17 ± 358.11	4.86	0.033 *
eTIV (mm^3^)	1,923,091.48 ± 145,178.78	1,987,143.36 ± 93,560.77	−0.64	0.526

Values are presented as the mean ± standard deviation (SD). All data were analyzed by covariance (ANCOVA) to compare hippocampal subfield volumes between CI and CN groups, accounting for sex, age, years of education and eTIV. Abbreviations: CA: Cornu Ammonis; GC-ML-DG: Molecular and Granule Cell Layers of the Dentate; HATA: Hippocampal–Amygdaloid Transition Area; eTIV: estimated Total Intracranial Volume. * *p* < 0.05.

**Table 3 brainsci-13-00460-t003:** Significant differences in white matter integrity between the cognitively normal group and cognitive impairment group.

Items	CN (21)	CI (25)	F	*p*-Value
CgH (MD)	0.00073 ± 0.000046	0.00077 ± 0.000051	4.24	0.046 *
CgH (AxD)	0.00115 ± 0.000062	0.00119 ± 0.000056	5.04	0.03 *

Values are presented as the mean ± standard deviation (SD). All data were analyzed by covariance (ANCOVA) to compare the WM integrity between CI and CN groups, accounting for sex, age and years of education. CN: cognitively normal; CI: cognitive impairment; CgH: Cingulum hippocampus; MD: mean diffusivity; AxD: Axial Diffusivity. * *p* < 0.05.

## Data Availability

If anyone is interested in extrapolating this data for further validation, the MRI images can be made available to the scientific community from the corresponding author.

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
