# Peer review of "Retinal Alterations as Potential Biomarkers of Structural Brain Changes in Alzheimer’s Disease Spectrum Patients"

_brainsci, 2023, doi:10.3390/brainsci13030460_

Round 1
Reviewer 1 Report
In this manuscript, authors have correlated the MRI and diffusion sensor based assessment of gray matter volume of the hippocampal subiculum/presuiculum or WM integrity of the CgH/UF in CI patients with retinal layer thickness and RFNL thickness measured with a OCT and retinal microvasculatures imaged with OCTA protocol. They found significant associations between retinal parameters and gray matter volume of the hippocampal subiculum/presuiculum or WM integrity CI patients. They claimed that suggested that the retinal layer thickness and microvascular measures acquired by OCTA could be potential markers for the early prediction of AD-related brain structural changes.
This is a nice piece of clinical work and manuscript is generally written well. However, authors are advised to incorporate the following comments/suggestions to enhance the qulaity and visibility of the paper.
1. Can authors provide a brief description, what mechanism causes the thickness reduction of the retina in AD patients ?
2. Does the retinal layer thickness changes is caused by ONL thicknesss reduction ?Could authors provide the the thickness measurements of outer nuclear layer (ONL) of retina in both healthy and patients data ?
3. Did authors find any changes in reflectivity of retinal layers ?
4. It would be great if authors can show the B-scans (cross-sectional images) of the retina from both healthy and patients eye.
5. I think it would be better to provide a general comment about OCT and OCTA technology to understand for the readers of broad range. OCT is an optical imaging method that allows both structural and functional imaging of retina of human as well as wide range of veterniary animals [1-3]. OCTA allow the extraction of blood vessels of retina from retinal OCT data. Authors are advised to provide the same/similar statements in the text with suggested good references (1. https://doi.org/10.1167/tvst.11.8.11 2. https://doi.org/10.1038/s41598-021-95320-z 3. https://doi.org/10.1186/s40942-015-0005-8)
Reviewer 2 Report
The study investigated an interesting topic of potential biomarkers of AD. Yet, there are critical methodological flaws that need to be addressed, especially in the method of segmenting the Hc subfields. Below I discussed the issues:
Method
1. participants: in the final 27 CI patients, how many were AD and how many were MCI?
2. For the paragraph of inclusion criteria, add references for all the cutoffs mentioned to justify the decisions. For example in line 113, why CDR score = 0 and ADL score = 8?
3. The information about the approach of assessing Hc subfields structures is not sufficient to assess its validity, and the validity of FS6.0 remains questionable (Wisse et al., 2021, HBM, https://pubmed.ncbi.nlm.nih.gov/33058385/) and proper quality control process and manual editing may be necessary (Samann et al):
a. T1-weighted images do not have the ideal contrast to visualize the internal landmarks for segmentation of the hippocampal subfields.
b. It is not clear what is the in-plane pixel size for the T1-weighted images, which is critical for accessing how the validity of the this method when segmenting the hippocampal subfields.
c. The validity of FreeSurfer 6.0’s hippocampal subfields segmentation against manual demarcation (which typically considered the gold rule if conducted by an expert of brain anatomy and text-retest reliability is established) remains unclear, and need to be established. Especially considering the current study deals with a special population (AD and MCI).
4. Normalized volume based on a ratio of raw volume and eTIV does not take into variance in individual difference in total brain volume – based on regression coefficient from a specific group. Also, the regression coefficients may be significantly different across groups of participants (AD vs MCI vs CN), then different coefficients may be used for different groups when correcting for individual differences in total brain volume.
There is a good amount of literature in the field comparing different ways of normalizing raw regional brain volumes. The authors may search and find the best way of normalization.
Results and conclusion:
5. How is the retinal layer thickness and microvascular measure different between CI and CN? Significantly different or not?
Writing:
6. English: first sentence in abstract has some grammar issue.
Reviewer 3 Report
The current study is trying to positively correlate retinal changes with significant brain alterations shown in cognitively impaired (CI) patients. For the above conclusion to be scientifically sound, some important considerations regarding the way the study was conducted need to be made.
Major comments:
• The only parameter of retinal measures found to be statistically different between the cognitively normal (CN) and cognitively impaired (CI) groups, is the Center Ring Vessel Density (Supp table 1). What does that indicate about the relationship between retinal parameters in general and cognitive impairment?
• Yet, the above parameter was never used to establish a possible correlation with the hippocampal subfields’ volume, the damaged fiber tracts and the white matter diffusivity (features that were compromised in CI group). Please explain why.
If it was used and no positive correlation was found (data not shown), please explain the possible negative result, i.e., why the only statistically different parameter (Center Ring Vessel Density) showed no positive correlation while other non-statistically different parameters did.
• Some selective parameters of retinal measures were shown to have positive correlation within the CI group. In order for this result to be used as a meaningful “marker” for cognitive impairment, one would have to show that such positive correlation does not exist within the CI group. Was this tested?
If not, why? If so, what were the results?
Minor comments:
Spelling and grammatical errors that need correction
• Line 15: change the beginning of the first sentence to:
….Retinal imaging being a potential biomarker for…..
• Line 43: change high to higher
• Lines 73-5: this is a repetition of the previous sentence
• Line 244: change higher to lower
